# OpenReview forum: "GSM8K-V: Can Vision Language Models Solve Grade School Math Word Problems in Visual Contexts"
_ICLR.cc/2026/Conference — ICLR 2026 Conference Desk Rejected Submission_

### Official Review · Reviewer_F6kb · 2025-10-27

**Soundness:** 4
**Presentation:** 4
**Contribution:** 4
**Rating:** 6
**Confidence:** 4

**Summary:**

This paper introduces GSM8K-V, a large-scale benchmark that converts the classic text-based GSM8K math dataset into a purely visual version. Each original problem is decomposed into scenes and translated into multiple images (average 4 per problem), forming 1,319 visually grounded math problems. The goal is to measure how well vision-language models (VLMs) can perform visual mathematical reasoning when all information is conveyed through images instead of text. Experiments show a strong modality gap—models that perform almost perfectly on text (e.g., Gemini-2.5-Pro with 95%) drop sharply on the visual version (around 47%), while humans remain highly accurate (≈91%). The authors also perform detailed error analysis, ablations, and comparisons of perception vs reasoning failures.

**Strengths:**

1. Builds the first visual equivalent of GSM8K, enabling a one-to-one comparison between text and visual reasoning tasks. Uses a three-stage generation process (problem decomposition → visual scene generation → human verification) to ensure semantic fidelity between text and image.

2.  Clearly exposes the visual reasoning gap of current VLMs, showing that perception and symbolic calculation remain unsolved under visual input.

3. Includes ablations on question format, number of images, and visual style, providing clear insight into what factors cause model failures.

4. Establishes a reliable human baseline and defines specific failure categories (e.g., perception-calculation error, instrument-reading error), offering actionable directions for model improvement.

**Weaknesses:**

1. The benchmark relies on a single, uniform “cartoon-style” visual domain for all problems. This limited diversity may cause models to overfit to specific visual cues instead of learning general, transferable visual reasoning skills. As a result, it remains unclear how well the benchmark performance reflects a model’s ability to handle more realistic visual formats such as photographs, sketches, or diagrams.

2. The current benchmark design mixes two different challenges — text recognition (OCR) and logical reasoning. Because many images contain numbers or symbols that must be read before reasoning, it is hard to separate perception errors from reasoning errors. A wrong answer may come from misreading a digit rather than misunderstanding the problem, making diagnosis and targeted improvement difficult.

3. Each textual problem is split into multiple visual scenes, forming a multi-image narrative input. This structure increases the demand for information integration and temporal consistency, but the paper does not analyze how this decomposition affects model performance. Without a controlled comparison between single-image and multi-image formats, it is unclear how much of the observed performance drop comes from multi-image reasoning complexity rather than from the core visual reasoning challenge itself.

**Questions:**

same as weakness

---

> ### Author Response · Authors · 2025-11-24
> **Rebuttal [1/2]**
>
> Dear reviewer F6kb,
>
> Thank you for your valuable feedback to help us improve our paper. We have revised our paper based on your feedback. We detail our response below and please kindly let us know if our response addresses your concerns.
>
> ---
> > **Q1:** **The benchmark relies on a single, uniform “cartoon-style” visual domain for all problems. This limited diversity may cause models to overfit to specific visual cues instead of learning general, transferable visual reasoning skills. As a result, it remains unclear how well the benchmark performance reflects a model’s ability to handle more realistic visual formats such as photographs, sketches, or diagrams.**
>
> **A1:** Thank you for raising this issue. We would like to clarify that we initially considered photorealistic rendering during benchmark design but ultimately opted for Pixar-style rendering to **avoid potential ethical, normative, and privacy risks associated with generating fully realistic imagery.** To specifically address the concern about the generalizability of cartoon-style illustrations, we conducted additional experiments using Real-World photorealistic rendering on the same subset (randomly sampled 10% of data) used in our style-ablation study.
> The results are shown in Table R1.
>
> **Table R1: Performance under different image rendering styles.**
>
> | **Model** | **Pixar Style** | **Giphli Style** | **Real-World Style** |
> | :---: | :---: | :---: | :---: |
> | GPT-4o | 27.27 | 28.93 | **29.70** |
> | GLM-4.5V | 28.10 | 26.45 | **25.74** |
> | Qwen2.5-VL-72B | 18.18 | 17.36 | **17.82** |
> | Qwen2.5-VL-7B | 12.40 | 12.40 | **11.88** |
>
> **The table shows that performance differences across all three styles are marginal** (within 0-3 percentage points), with no
> systematic degradation toward realism. This confirms that **stylistic variation minimally impacts reasoning and demonstrates robustness to photorealistic rendering, validating cross-style generalization**. We have revised the paper and put these
>  details in Section 5.3 and Appendix B.5.

---

> ### Author Response · Authors · 2025-11-24
> **Rebuttal [2/2]**
>
> > **Q2:** **The current benchmark design mixes two different challenges — text recognition (OCR) and logical reasoning. Because many images contain numbers or symbols that must be read before reasoning, it is hard to separate perception errors from reasoning errors. A wrong answer may come from misreading a digit rather than misunderstanding the problem, making diagnosis and targeted improvement difficult.**
>
> **A2:** Thank you for your question. The concern about mixing perception and reasoning failures is valid. To address this, we conducted a detailed error analysis to clearly separate different failure sources rather than blending them. Based on manual examination of model outputs, we classify errors into four diagnostic categories:
>
> *   **Perception Calculation Error:** Miscounting objects or confusing categories leads to incorrect numerical grounding and subsequent calculation mistakes.
>
> *   **Implicit Visual Inference Error:** Key relations appear implicitly in visual context, but models often treat them as irrelevant details, failing to form coherent relational reasoning.
>
> *   **Instrument-Reading Error:** Numerical values are encoded in everyday measurement instruments, and models sometimes misinterpret them.
>
> *   **Other Error:** Includes cases mainly related to OCR-type character mistakes and general visual confusion.
>
>
> Using these categories, we analyze Qwen2.5-VL-72B and GPT-4o. The dominant sources of failure are Perception Calculation Error and Implicit Visual Inference Error, accounting for 34% and 28% of errors in Qwen2.5-VL-72B, and 30% and 26% in GPT-4o. **This result shows that the major challenge does not come from OCR but from visual perception and visual implicit relational reasoning.** We have revised the paper and put these details in Section 5.5 and Appendix E.
>
> > **Q3:** **Each textual problem is split into multiple visual scenes, forming a multi-image  narrative input.  This structure  increases the demand for information integration and temporal consistency, but the paper does not analyze how this decomposition affects model performance. Without a controlled comparison between single-image and multi-image formats, it is unclear how much of the observed performance drop comes from multi-image reasoning complexity rather than from the  core visual reasoning challenge itself.**
>
> **A3:** Thank you for your question. To examine whether performance is influenced by scene organization, we evaluate four controlled settings: Single-Image, Single-Image (r\*), Multi-Image, and Multi-Image (r\*), where r\* represents a random permutation of scene order that is applied consistently across both formats. The comparison is reported in Table R2.
>
> **Table R2: Comparison under single-image and multi-image input formats. r\* indicates that images are presented in a randomized order.**
>
> | **Model** | **Single-Image** | **Single-Image (r\*)** | **Multi-Image** | **Multi-Image (r\*)** |
> | --- | :---: | :---: | :---: | :---: |
> | GPT-4o | 27.45 | **25.85** | 29.57 | **26.54** |
> | GLM-4.5V | 26.76 | **24.29** | 28.28 | **26.91** |
> | Qwen2.5-VL-72B | 20.47 | **18.35** | 21.91 | **18.12** |
> | Qwen2.5-VL-7B | 15.39 | **5.16** | 15.16 | **3.87** |
>
> Across all models, randomized ordering leads to a performance decline. GPT-4o, GLM-4.5V, and Qwen2.5-VL-72B show only modest drops, indicating that they are already challenged by cases where key information is implicitly distributed across scenes. In contrast, Qwen2.5-VL-7B experiences a sharp degradation and often claims that the clues are insufficient, suggesting difficulty in accumulating evidence once the scene structure is disrupted. **These results indicate that weaker models are sensitive to scene organization, while stronger models are limited mainly by the challenge of interpreting visually embedded semantics. The multi-image format adds integration demands but does not account for the primary performance gap in GSM8K-V.**
>
> ---
>
> Thank you again for the thorough review. We think that our rebuttal can address your main concern, if you have any further questions, we would be happy to answer them.
>
> Sincerely,
> GSM8K-V Authors

---

### Official Review · Reviewer_GZam · 2025-10-28

**Soundness:** 2
**Presentation:** 3
**Contribution:** 2
**Rating:** 2
**Confidence:** 5

**Summary:**

This paper introduces GSM8K-V, a purely visual and multi-image benchmark for mathematical reasoning. The authors systematically transform each sample from the text-based GSM8K dataset into visual form through an automated image-generation and human-annotation pipeline, producing 1,319 high-quality samples. The work highlights the limitations of current VLMs in grounding mathematical reasoning in visual contexts.

**Strengths:**

The work addresses an important gap by extending mathematical reasoning evaluation beyond geometric diagrams to multi-image visual word problems.

The mapping from textual GSM8K samples to visual counterparts with a well-defined image-generation and validation pipeline that combines automation with human verification.

The authors benchmark a broad set of open- and closed-source VLMs.

**Weaknesses:**

The paper lacks a clear rationale for converting text-based math word problems into purely visual, multi-frame representations. In current multimodal large language model design, the canonical two-stage process involves visual perception for diagram understanding followed by symbolic reasoning in the language model — leveraging both modalities. Completely removing textual input seems counterintuitive and does not necessarily yield a more realistic or informative evaluation compared to existing benchmarks such as MathVerse or MathVista.

In terms of realism perspective, short textual prompts (like fewer than 50 tokens) into several images may introduce artificial visual complexity rather than emulate real-world reasoning. If realism is the goal, video-based datasets such as VideoMathQA (VideoMathQA: Benchmarking Mathematical Reasoning via Multimodal Understanding in Videos) provide a more natural form of multimodal reasoning involving temporal perception and sequential inference from the real world.

The benchmark closely resembles OCR-based mathematical reasoning, where models must parse textual information embedded in images. The paper does not adequately clarify how GSM8K-V differs from existing OCR or document-understanding benchmarks. Including baselines models such as MinerU or PaddleOCR would help quantify this distinction.

The contribution is largely restricted to dataset construction. The paper would benefit from demonstrating how GSM8K-V design pipeline/workflow could be used to design specific training signals or losses that enhance reasoning over visual information, and showing generalization from GSM8K-V-trained models to real-world multimodal data (e.g., whiteboard videos, scanned worksheets) would strengthen both novelty and applicability.

**Questions:**

The work does not clearly articulate what unique cognitive or reasoning abilities GSM8K-V tests compared to existing benchmarks. I agree with “one picture is worth a thousand words” but this paper is inverted here: concise textual expressions are expanded into multi-image sequences. The authors may consider how to  clarify what new aspects of multimodal reasoning this transformation reveals, or what insights it offers for developing future MLLMs.

---

> ### Author Response · Authors · 2025-11-24
> **Rebuttal [1/4]**
>
> Dear reviewer GZam,
>
> Thank you for your valuable feedback to help us improve our paper. We have revised our paper based on your feedback. We detail our response below and please kindly let us know if our response addresses your concerns.
>
> ---
> > **Q1:** **The paper lacks a clear rationale for converting text-based math word problems into purely visual, multi-frame representations. In current multimodal large language model design, the canonical two-stage process involves visual perception for diagram understanding followed by symbolic reasoning in the language model — leveraging both modalities. Completely removing textual input seems counterintuitive and does not necessarily yield a more realistic or informative evaluation compared to existing benchmarks such as MathVerse or MathVista.**
> **The work does not clearly articulate what unique cognitive or reasoning abilities GSM8K-V tests compared to existing benchmarks.**
>
> **A1:** Thank you for your question. GSM8K-V builds on the nature of GSM8K: the original problems are everyday math scenarios, and the text is only an abstraction of real situations. GSM8K-V restores these problems to visual scenes, reflecting how such reasoning typically appears in daily life. The goal is not to make the task harder by removing text, but to see whether models can solve the same problems when they arise visually in daily life, as humans often encounter them.
>
> Existing benchmarks \[1-5\] focus on structured diagrams or geometry-based layouts. Their reasoning process relies on recognizing visual content and then converting it into symbolic text for calculation. GSM8K-V evaluates a different capability — whether a model can interpret information from multi visual scenes in daily life. The detailed comparision are shown in Table R1.
>
> **Table R1: Comparison of visual mathematicl benchmarks**
>
> | Benchmark | Input | Evaluated Abilities | Target Scenario |
> | --- | --- | --- | --- |
> | MathVista \[1\] | Single-image | Geometry & Algebra, Perception-based reasoning | Classroom-style visual math problems |
> | MathVision \[3-4\] | Single-image\* | Geometry, Diagram understanding, Logical reasoning | Classroom-style structured math tasks |
> | MathVerse \[2\] | Single-image | Symbolic reasoning after visual perception, Diagram interpretation | Classroom-style math-worksheet problems |
> | VideoMathQA \[5\] | Video | Temporal reasoning, Cross-frame perception, Elementary logical reasoning | Instructional video-based math tasks |
> | **GSM8K-V (Ours)** | **Multi-image** | **Perception & Counting, Visual implicit reasoning, Everyday logical reasoning** | **Real-world math word problems in daily life** |
>
> As Table R1 summarizes, GSM8K-V uniquely focuses on real-world visual reasoning, whereas existing benchmarks remain centered on classroom-style mathematical tasks. The visual implicit reasoning (**information is often conveyed through transitions across multiple visual scenes**), and everyday logical reasoning is unique and different than other benchmark.
>
> Ref:
>
> \[1\] lu et al, MathVista: Evaluating mathematical reasoning of foundation models in visual contexts
>
> \[2\] zhang et al, MathVerse: Does Your Multi-modal LLM Truly See the Diagrams in Visual Math Problems
>
> \[3\] wang et al, Measuring Multimodal Mathematical Reasoning with MATH-Vision Dataset
>
> \[4\] wang et al, MathCoder-VL: Bridging Vision and Code for Enhanced Multimodal Mathematical Reasoning
>
> \[5\] rasheed et al, VideoMathQA: Benchmarking Mathematical Reasoning via Multimodal Understanding in Videos
>
> > **Q2:** **Weakness: In terms of realism perspective, short textual prompts (like fewer than 50 tokens) into several images may introduce artificial visual complexity rather than emulate real-world reasoning. If realism is the goal, video-based datasets such as VideoMathQA (VideoMathQA: Benchmarking Mathematical Reasoning via Multimodal Understanding in Videos) provide a more natural form of multimodal reasoning involving temporal perception and sequential inference from the real world.**
>
> **A2:** Thank you for your question. Current video-based math benchmarks, including VideoMathQA \[1\], focus on instructional or educational settings, and Video-MMMU \[2\], focus on subject education scene. However, there is still no benchmark targeting everyday visual reasoning in video form. GSM8K-V serves as a first step in this direction. We begin with GSM8K, whose problems are rooted in real-world situations, and convert them into multi-image visual scenarios. Our intention is not to increase complexity artificially, but to approximate how such problems naturally arise outside text in daily life. We hope future work will lead to video-based benchmarks that reflect everyday scenarios more fully.
>
> Ref:
>
> \[1\] rasheed et al, VideoMathQA: Benchmarking Mathematical Reasoning via Multimodal Understanding in Videos
>
> \[2\] hu et al, Video-MMMU: Evaluating Knowledge Acquisition from Multi-Discipline Professional Videos

---

> ### Author Response · Authors · 2025-11-24
> **Rebuttal [2/4]**
>
> > **Q3:** **The benchmark closely resembles OCR\-based mathematical reasoning, where models must parse textual information embedded in images. The paper does not adequately clarify how GSM8K\-V differs from existing OCR or document\-understanding benchmarks. Including baselines models such as MinerU or PaddleOCR would help quantify this distinction.**
>
> **A3:** We appreciate your suggestion and agree that the distinction from OCR\-based reasoning should be clearly demonstrated. In GSM8K\-V, OCR is intentionally insufficient for solving the problems. We first used Qwen\-OCR to extract all textual elements from each image and used the extracted content as input to other models. The performance dropped substantially compared to image\-only, showing that simple text parsing cannot recover the required semantic cues.
>
> To further strengthen this point and integrate your valuable suggestion, we additionally tested input settings (OCR combined with the original image) and MinerU2.5 \[1\], PaddleOCR\-VL \[2\] baseline. The results are presented in Table R2.
>
> **Table R2: Performance Comparison about ocr input settings**
> | Model | Text only | Imag only | Qwen-OCR | **PaddleOCR-VL** | **MinerU2.5** | **Qwen-OCR + Image** | **PaddleOCR-VL + Image** | **MinerU2.5 + Image** |
> | --- | :---: | :---: | :---: | :---: | :---: | :---: | :---: | :---: |
> | GPT-4o | 94.92 | 29.57 | 11.98 | **12.05** | **12.13** | **33.06** | **39.50** | **39.12** |
> | GLM-4.5V | 90.60 | 28.28 | 9.33 | **10.24** | **10.01** | **32.22** | **37.98** | **37.91** |
> | Qwen2.5-VL-72B | 94.09 | 21.91 | 9.63 | **10.08** | **10.01** | **24.34** | **27.82** | **27.82** |
> | Qwen2.5-VL-7B | 78.32 | 15.16 | 3.11 | **4.17** | **4.02** | **16.45** | **17.97** | **17.74** |
>
>
> From Table R2, we observe that when models are given only the outputs of MinerU2.5 or PaddleOCR\-VL, their accuracy remains very low. This indicates that **GSM8K\-V cannot be solved via text recognition** alone and necessarily requires implicit visual reasoning. Adding the original image to the OCR results yields some improvement, but the overall performance is still limited. This shows that **OCR capability alone is insufficient, and the small accuracy gain further indicates that additional types of errors are involved beyond text recognition.** We have revised the paper and put these details in Appendix B.3.
>
> Ref:
>
> \[1\] niu et al, MinerU2.5: A Decoupled Vision\-Language Model for Efficient High\-Resolution Document Parsing
>
> \[2\] cui et al, PaddleOCR\-VL: Boosting Multilingual Document Parsing via a 0.9B Ultra\-Compact Vision\-Language Model

---

> ### Author Response · Authors · 2025-11-24
> **Rebuttal [3/4]**
>
> >**Q4:** **The contribution is largely restricted to dataset construction. The paper would benefit from demonstrating how GSM8K\-V design pipeline/workflow could be used to design specific training signals or losses that enhance reasoning over visual information, and showing generalization from GSM8K\-V-trained models to real-world multimodal data (e.g., whiteboard videos, scanned worksheets) would strengthen both novelty and applicability.**
> **The authors may consider how to clarify what new aspects of multimodal reasoning this transformation reveals, or what insights it offers for developing future MLLMs.**
>
> Thank you for your question. Since the final weakness and question are closely related, we address them together.
>
> 1.  **Regard contribution, novelty and applicability**
>
>     Unlike mainstream math benchmarks, GSM8K\-V aims to reveal the limitations of current VLMs in everyday visual reasoning, rather than structured diagrams or symbolic layouts. To verify this point, we conduct performance comparion experiment based on VL-Rethinker-7B \[2\].
>
>     **Table R3: Performance comparison between Qwen2.5-VL-7B-Instruct and VL-Rethinker-7B across multiple benchmarks.**
>
>     | Model | MathVista (testmini) | MathVerse (testmini) | MathVision (test) | GSM8K\-V |
>     | --- | :---: | :---: | :---: | :---: |
>     | Qwen2.5-VL-7B-Instruct \[1\] | 68.2 | 46.3 | 25.1 | 15.16 |
>     | VL-Rethinker-7B \[2\] | 74.9 | 54.2 | 32.3 | 14.56 |
>     | **Δ** | **+6.7** | **+7.9** | **+7.2** | **\-0.6** |
>
>     As shown in Table R3, VL-Rethinker-7B is derived from Qwen2.5-VL-7B-Instruct and further trained on ViRL39K \[2\] (38,870 verifiable QAs for vlm RL training, ~ 80% math type).  From Table R3, RL training on ViRL39K significantly improves performance on MathVista, MathVerse, MathVision(+6.7, +7.9, +7.2). However, performance drops on GSM8K\-V (−0.6). This result indicates that GSM8K\-V represents a distinct task type rather than an extension of existing benchmarks.
>
> 2.  **Aspects of multimodal reasoning**
>
>     GSM8K\-V evaluates whether a model can interpret information from multi visual scenes in daily life. This task mainly includes these following reasoning aspects:
>
>     *   Perception and counting in daily life scenes
>
>     *   Visual implicit reasoning and Cross-scene integration
>
>     *   Visual logical reasoning

---

> ### Author Response · Authors · 2025-11-24
> **Rebuttal [4/4]**
>
> 3.   **Insights for developing future MLLMs and specific training signals or losses**
>
>
>     At present,mainstream multimodal systems typically follow a two-stage reasoning process \[3\] :  visual perception first, followed by symbolic reasoning inside the language model.  We know that perception has long been a weakness of multimodal models \[4\], yet in many cases the model does not merely fail to see clearly but rather fails to understand what it sees, especially when information is expressed implicitly in daily life. If these signals are not interpreted correctly, downstream reasoning naturally breaks down, which is consistent with the observations emphasized by GSM8K-V.
>
>     Based on this insight, model capability can be strengthened from both data and training objectives. **On the data side, improving a model’s ability to handle implicit visual expression requires high-quality data that share a central property**: key information is not given explicitly through text or symbols. Instead, it is partially and sparsely embedded across different visual elements. With such data, the model cannot rely on captions or OCR to extract answers. It must learn to reconstruct relations, detect changes, recover quantities and understand scene logic under conditions, where visual observations are incomplete and not explicitly stated.
>
>     Implicit visual expression therefore introduces a new supervision dimension. It can be used as an action signal that encourages the model to internalize the underlying visual operations. On the training-objective side\[5\], following RLVR, one can define rewards that target visual reasoning rather than only final correctness. For example, **rewards can encourage accurate recognition of visual state changes or the production of intermediate reasoning steps that remain faithful to the visual facts**. These signals can be naturally incorporated into GRPO or its variants, extending RL optimization from a focus on outcome correctness to a focus on the correctness of the visual understanding process itself.
>
>     Ref:
>
>     \[1\] Qwen Team, Qwen2.5-VL Technical Report
>
>     \[2\] wang et al, VL-Rethinker: Incentivizing Self-Reflection of Vision-Language Models with Reinforcement Learning
>
>     \[3\] Yang et al, R1-Onevision: Advancing Generalized Multimodal Reasoning through Cross-Modal Formalization
>
>     \[4\] Bigverdi et al, Perception Tokens Enhance Visual Reasoning in Multimodal Language Models
>
>     \[5\] DeepSeekTeam, DeepSeekMath: Pushing the Limits of Mathematical Reasoning in Open Language Models
>
> ---
>
> Thank you again for the thorough review. We think that our rebuttal can address your main concern, if you have any further questions, we would be happy to answer them.
>
> Sincerely,
> GSM8K-V Authors

---

> > ### Comment · Reviewer_GZam · 2025-11-25
> >
> > Thank you for the authors’ rebuttal and for conducting the additional experiments on OCR-based models. I appreciate the effort to clarify some of the concerns. However, my core questions remain addressed.
> >
> > First, I still do not fully agree or understand the motivation or value of translating textual corpora into visual-only animations. There already exist numerous real-world, vision-centric benchmarks, so it remains unclear why such a translation is necessary. For tasks that rely on visual perception, such as maze navigation, Sudoku, and pattern-matching benchmarks, like the Abstraction and Reasoning Corpus (ARC).  In the rebuttal, the authors argue that GSM8k-V aims to target real-world, everyday visual reasoning, comparing to VideoMathQA. This is a very strong claim and, at present, it seems overstated. I encourage the authors to provide clearer definition or evidence supporting the terms “everyday” or “daily-life visual reasoning,”  demonstrating how these claims are grounded in the dataset design and reflected in the benchmark distribution.
> >
> > Second, the authors also state that GSM8k-V focuses on perception, but in Fig. 3 I do not see perception-oriented tasks, like shape classification, detection, or grounding tasks. Most of the tasks shown appear to be logical in nature, so what the perception authors indicate?
> >
> > Third, the authors mention that “performance dropped substantially compared to image-only, showing that simple text parsing cannot recover semantic cues.” This raises an additional question: what is the primary target of this work? for example, in domain of geometric problem solving, the central challenge is to perceive geometric primitives, understand their spatial relations, and perform visually grounded reasoning, rather than semantic scene understanding. These geometric primitives are inherently symbolic and typically appear in semantic-sparse, schematic diagrams. So, I still do not see the relevance or necessity of introducing too much “semantic scenarios” into the animations.

---

> > > ### Author Response · Authors · 2025-11-26
> > > **Response [1/3]**
> > >
> > > Dear reviewer GZam,
> > >
> > > Thank you once again for your feedback and follow-up question. We detail our response below and please kindly let us know if our response addresses your concerns.
> > >
> > > > **Q1: First, I still do not fully agree or understand the motivation or value of translating textual corpora into visual-only animations. There already exist numerous real-world, vision-centric benchmarks, so it remains unclear why such a translation is necessary. For tasks that rely on visual perception, such as maze navigation, Sudoku, and pattern-matching benchmarks, like the Abstraction and Reasoning Corpus (ARC). In the rebuttal, the authors argue that GSM8k-V aims to target real-world, everyday visual reasoning, comparing to VideoMathQA. This is a very strong claim and, at present, it seems overstated. I encourage the authors to provide clearer definition or evidence supporting the terms “everyday” or “daily-life visual reasoning,” demonstrating how these claims are grounded in the dataset design and reflected in the benchmark distribution.**
> > >
> > >
> > >
> > > [A1-1]: **The motivation of GSM8K-V is not merely to translate textual benchmarks into visual form**, but rather to **construct a dataset that evaluates mathematical reasoning as it naturally occurs in everyday visual contexts**. The transformation from text to visual scenes is **a method**, not the motivation itself. GSM8K naturally provides a base for constructing everyday visual reasoning tasks. The conversion to visual format enables *controlled comparison* between **text-based reasoning vs. visual-based reasoning** under the **same semantic content**, allowing us to isolate whether failures come from:
> > >
> > > * inability in visual perception
> > > * inability in visual implicit reasoning
> > > * or insufficient mathematical reasoning ability
> > >
> > > While current benchmarks (e.g., MathVista, VideoMathQA) **rarely** evaluate whether models can extract mathematically relevant information directly from everyday visual scenes, where key cues are implicitly embedded in natural contexts(i.e., everyday visual reasoning). They primarily focus on domain-specific skills and structured knowledge, rather than everyday reasoning found in real-life situations.
> > > We have revised the paper and provided detailed error analyses in Section 5.5 and Appendix E.

---

> > > > ### Author Response · Authors · 2025-11-26
> > > > **Response [2/3]**
> > > >
> > > > [A1-2]: We describe ***everyday visual reasoning in GSM8K-V*** as: **The ability to interpret information distributed across real-world visual scenes and integrate it with mathematical reasoning, without relying on domain-specific visual skills or explicit symbolic cues.**
> > > >
> > > > It does not test pattern-oriented or rule-based visual skills (e.g., Sudoku, ARC, diagram puzzles) [1-2], but rather evaluates whether models can apply mathematical reasoning in contexts that resemble everyday visual experience. Consider a simplified example(details in Appendix E.2):
> > > >
> > > > > Textual Problem: Boris has 11 apples. Beck has 5 apples at first, but she ate 2 apples. If Boris gives Beck 3 apples, how many fewer apples does Beck have than Boris now?
> > > >
> > > >
> > > > - Visual Scene 1: 11 apples are on the wooden table, Boris looks and points at his apples.
> > > >
> > > > - Visual Scene 2: 5 apples are on the silver table, Beck looks and points at her apples.
> > > >
> > > > - Visual Scene 3: Only 3 apples remain on the silver table.
> > > >
> > > > - Visual Scene 4: Boris gives Beck 3 apples.
> > > >
> > > > Table R1 further summarizes the key abilities that GSM8K-V aims to evaluate as part of everyday visual reasoning.
> > > >
> > > >
> > > > **Table R1: Key Characteristics of Everyday Visual Reasoning in GSM8K-V**
> > > >
> > > > | Characteristic                  | Description                                                                                     | Realistic Example                                                                                                                                     |
> > > > | ------------------------------- | ----------------------------------------------------------------------------------------------- | ----------------------------------------------------------------------------------------------------------------------------------------------------- |
> > > > | **perception & Counting** | The model needs to jointly use object recognition, spatial perception, and numerical reasoning. | A shopping cart contains ten apples, five bananas, and six oranges. To answer "How many apples are there?", the model must visually distinguish categories and count only the relevant objects.                      |
> > > > | **Cross-scene integration**          | Key information is distributed across multiple visual scenes and must be integrated to reconstruct reasoning steps. | Former Example: From Scene 1 (11 apples) → Scene 2 (5 apples) → Scene 3 (3 apples) → Scene 4 (apple transfer), the model must integrate scene transitions to infer changes in quantity. |
> > > > | **Partial observability**            | Some intermediate states are not visible, yet must still be inferred to complete the reasoning chain.               | Former Example: The act of eating apples is never shown explicitly — Scene 2 and Scene 3 imply that Beck must have eaten 2 apples, even though that action is not displayed.            |
> > > > | **Contextual reasoning**             | Real-world knowledge must be applied to interpret visual states and actions.                                        | Former Example: The action "giving apples" is inferred from visual cues (Boris’s gesture and movement) rather than an explicit textual statement of transfer.                           |
> > > > | **Non-domain-specific**         | Not based on puzzle-solving rules (e.g., Sudoku, ARC [1-2]), but everyday logic.                      | Various examples in daily life requiring everyday visual reasoning.                 |
> > > >
> > > > Noting: the cross-scene integration, partial observability and contextual reasoning characteristics are included in our previous state（i.e., **Implicit visual reasoning**）
> > > >
> > > >
> > > > [1] Chollet et al. ARC-AGI-2: A New Challenge for Frontier AI Reasoning Systems
> > > >
> > > > [2] Seely et al. Sudoku-Bench: Evaluating creative reasoning with Sudoku variants

---

> > > > > ### Author Response · Authors · 2025-11-26
> > > > > **Response [3/3]**
> > > > >
> > > > > >**Q2: Second, the authors also state that GSM8k-V focuses on perception, but in Fig. 3 I do not see perception-oriented tasks, like shape classification, detection, or grounding tasks. Most of the tasks shown appear to be logical in nature, so what the perception authors indicate?**
> > > > > >
> > > > > [A2]: Thank you for pointing out this question. **The distribution in Fig. 3 is not defined by ability types**, but rather by **real-world daily-life contexts** where mathematical reasoning occurs. However, solving these tasks still requires perception before reasoning can begin. We examplify some category tasks in Table R2 and provided detailed case studies for each category in Appendix F in paper.
> > > > >
> > > > >
> > > > > **Table R2: Different category examples**
> > > > > | Category (Fig. 3)                        | Role of Visual Perception                                                             | Example                                                                                                                                                                      |
> > > > > | ---------------------------------------- | ------------------ | -------------------------------------------------------------------------------- |
> > > > > | **Measurement (Length / Area / Volume)** | The model must *recognize* different objects (e.g., sofa vs. rug) and *compare* their relative sizes visually before applying textual reasoning. | A living room scene shows a chair, rug, and couch with different dimensions and length relation in spatial arrangement.                      |
> > > > > | **Signboard & Icon (Price)**             | Perception is required to *detect price tags, symbols, and shop signs*, grounding them to semantic meaning before numerical reasoning is possible.          | A price list in the bookstore shows that differnent books icon and its price tag ($18, $13, $8).                    |
> > > > > | **Temporal (Clock)**                     | The model must *read clock faces, locate hour and minute hands,* and associate them with daily routines before temporal reasoning is possible.              | A clock is shown with an alarm symbol in the morning and yoga equipment at night.                |
> > > > >
> > > > >
> > > > >
> > > > > > **Q3: Third, the authors mention that “performance dropped substantially compared to image-only, showing that simple text parsing cannot recover semantic cues.” This raises an additional question: what is the primary target of this work? for example, in domain of geometric problem solving, the central challenge is to perceive geometric primitives, understand their spatial relations, and perform visually grounded reasoning, rather than semantic scene understanding. These geometric primitives are inherently symbolic and typically appear in semantic-sparse, schematic diagrams. So, I still do not see the relevance or necessity of introducing too much “semantic scenarios” into the animations.**
> > > > >
> > > > >
> > > > >
> > > > > [A3]: We agree that geometric benchmarks play a fundamental role in evaluating symbolic reasoning on schematic diagrams. In real-life situations, mathematical reasoning seldom appears in purely symbolic or diagrammatic form. Instead, quantities are grounded in **semantic visual states**—objects may be partially hidden, changed across time, or implicitly modified by actions (e.g., eating, giving, moving). As we state in A1, GSM8K-V explicitly tests **whether models can understand what is happening in the everyday visual scene** before applying mathematics.
> > > > >
> > > > > This explains why semantic content in the visual scenes is not noise, but **a necessary component** for assessing everyday visual reasoning. To validate this, we conducted OCR-only and OCR+image experiments: if simple text parsing could recover semantic cues, accuracy should be high. However, accuracy dropped substantially (Table R2 in rebuttal), demonstrating that the required information is **not text-extractable** and instead must be visually understood. This indicates that GSM8K-V evaluates a different capability than geometry-based benchmarks. The comparison is shown in Table R3.
> > > > >
> > > > >
> > > > > **Table R3: Benchmark comparison**
> > > > > | Type of Reasoning                     | What it Evaluates              | Representation          | Required Before Reasoning                                         |
> > > > > | ---------------------- | ----------------- | ----------------------- | ----------------------- |
> > > > > | Geometric reasoning (e.g., MathVista) | Symbolic spatial relations                                         | Clean diagrams          | Parsing geometric primitives                                      |
> > > > > | **GSM8K-V (ours)**                    | Understanding current visual state & extracting math-relevant cues | Natural semantic scenes | Perceiving actions / objects / quantities **from visual context** |
> > > > >
> > > > > ---
> > > > > ---
> > > > >
> > > > > Thank you again for your follow-up and feedback. We think that our reply can address your main concern, if you have any further questions, we would be happy to answer them.
> > > > >
> > > > > Sincerely,
> > > > > GSM8K-V Authors

---

### Official Review · Reviewer_bjKz · 2025-10-31

**Soundness:** 4
**Presentation:** 4
**Contribution:** 3
**Rating:** 4
**Confidence:** 3

**Summary:**

This paper builds upon the well-known benchmark GSM8K (grade school math problems) and presents GSM8K-V by converting ~1.3K text-only questions to their purely-visual comic-based counterparts. The best model, Gemini pro's performance on the new benchmark drops drastically from the original performance of ~95% to ~47%, demonstrating the headroom the new benchmark introduces to the research community.

**Strengths:**

1- The paper is quite interesting and nice to read, and the idea is novel.

2- The benchmark is new and challenging and has meticulous human annotations as mentioned by the paper. This is quite a contribution to the community.

3- Extensive results, sensitivity/ablation/error analyses as well as detailed evaluations, make it a whole lot easy to reproduce the findings.

**Weaknesses:**

1- The pipeline works extensively and heavily based on closed-source models such as GPT and hence is expensive (as mentioned in Table 7).

2- Pixar and Giphli styles show minimum differences using ablation study, but it's not clear if the observations made in the paper can still be made using noisy and real-world pictures.

3- Table 5 is somehow confusing, as I can't wrap my head around the logic behind the proof. For instance, GPT-4o's performance degrades badly using OCR but with caption achieves 55% which is still better than image-only ~29%. This may suggest the model is simply not good enough at perceiving images rather than being a modality issue.

**Questions:**

1- What would happen to the models performances if the images are presented in real-world format rather than cartoons?

2- As also mentioned in the last weakness, I still don't understand why captions are better than images? It is because the structured text in the caption helped?

---

> ### Author Response · Authors · 2025-11-24
> **Rebuttal [1/2]**
>
> Dear Reviewer bjKz,
>
> Thank you for your valuable feedback to help us improve our paper. We have revised our paper based on your feedback. We detail our response below and please kindly let us know if our response addresses your concerns.
>
> ---
> > **Q1:** **The pipeline works extensively and heavily based on closed-source models such as GPT and hence is expensive (as mentioned in Table 7).**
>
> **A1:** Thank you for your question. Our decision to use closed-source models follows extensive pilot testing. GPT-4.1 was  selected because it consistently produced logically
> coherent multi-scene descriptions with correct quantitative transitions, which is essential for encoding GSM8K-style problems
> into visual form. GPT-Image-1 was chosen because it showed the  most reliable cross-scene
> consistency and stable preservation of characters, attributes, and object states. By contrast, open-source generators
> frequently introduced drift across scenes or altered key entities, which would compromise benchmark validity. **Although this choice increases construction cost, it is essential for producing a high-quality benchmark(semantically  accurate, visually consistent) to ensure the the integrity of downstream evaluation.**
>
> > **Q2:** **Pixar and Giphli styles show minimum differences using ablation study, but it's not clear if the observations made in the paper can still be made using noisy and real-world pictures.**
> **What would happen to the models performances if the images are presented in real-world format rather than cartoons?**
>
> **A2:** Thank you for your question. We would like to clarify that we initially considered photorealistic rendering during benchmark design but ultimately opted for Pixar-style rendering to **avoid potential ethical, normative, and privacy risks associated  with generating fully realistic imagery.** To specifically address the concern about the generalizability of cartoon-style  illustrations, we conducted additional experiments using Real-World photorealistic rendering on the same subset (randomly  sampled 10% of data) used in our style-ablation study. The results are shown in Table R1.
>
> **Table R1: Performance under different image rendering styles.**
>
> | **Model** | **Pixar Style** | **Giphli Style** | **Real-World Style** |
> | :--- | :---: | :---: | :---: |
> | GPT-4o | 27.27 | 28.93 | **29.70** |
> | GLM-4.5V | 28.10 | 26.45 | **25.74** |
> | Qwen2.5-VL-72B | 18.18 | 17.36 | **17.82** |
> | Qwen2.5-VL-7B | 12.40 | 12.40 | **11.88** |
>
> **The table shows that performance differences across all three styles are marginal** (within 0-3 percentage points), with no  systematic degradation toward realism. This confirms that **stylistic variation minimally impacts reasoning and demonstrates robustness to photorealistic rendering, validating cross-style generalization**. We have revised the paper and put these details  in Section 5.3 and Appendix B.5.

---

> ### Author Response · Authors · 2025-11-24
> **Rebuttal [2/2]**
>
> >**Q3:** **Table 5 is somehow confusing, as I can't wrap my head around the logic behind the proof. For instance, GPT4o's performance degrades badly using OCR but with caption achieves 55% which is still better than image-only ~29%.**
> **This may suggest the model is simply not good enough at perceiving images rather than being a modality issue. As also mentioned in the last weakness, I still don't understand why captions are better than images? It is because the structured text in the caption helped?**
>
> **A3:** Thank you for your question. The caption setting in Table 5 does not rely on an auxiliary captioning model. Instead, it directly uses the Stage-2 scene descriptions that were produced during dataset construction to synthesize each image**. Feeding these captions into a VLM effectively bypasses the visual-perception stage and exposes the model to the most accurate symbolic summary available, which naturally leads to higher accuracy than the image-only setting.** OCR transcriptions (via Qwen-OCR) lose the essential visual cues required for solving GSM8K-V tasks. They do not capture quantities, spatial relations, object persistence, or visual transitions across scenes. Their consistently low performance across all models confirms that GSM8K-V cannot be solved by text extraction alone and indeed requires genuine visual reasoning.
>
> *   Text-only: all information are fully explicit in symbolic form (i.e., original gsm8k original textual question).
>
> *   Image-only: information is encoded implicitly through visual cues such as spatial layout, object states, and changes across scenes (i.e., our GSM8K-V).
>
> *   OCR: only surface text is extracted (via Qwen-OCR), so most semantic cues are missing and essential visual information is lost.
>
> *   Caption: uses the Stage-2 scene descriptions from dataset construction, following the tripartite schema (object, action, composition).
>
>
> However, the performance improvement is much smaller than one would intuitively expect, and caption accuracy remains far below text-only performance. **This gap can be traced to the structural properties of the captions, which follow the tripartite schema (object, action, composition) introduced in Section 3.2 and are primarily optimized for image synthesis rather than semantic clarity. As a result, key semantic is often embedded implicitly across the described elements rather than directly expressed.** This behavior reflects a broader distinction between visual and textual modalities. Such representational differences contribute to the remaining performance gap between caption-based and text-only settings. We have revised the paper and put these details in Section 5.4 and Appendix B.3.
>
> ---
>
> Thank you again for the thorough review. We think that our rebuttal can address your main concern, if you have any further questions, we would be happy to answer them.
>
> Sincerely,
> GSM8K-V Authors

---

> > ### Comment · Reviewer_bjKz · 2025-11-25
> > **Increase score**
> >
> > I'd like to thank authors for carefully addressing my concerns, and as I said in my original review this is a fantastic research and deserves to be published. I'll increase my score and vote in favor of the acceptance.

---

> > > ### Author Response · Authors · 2025-11-26
> > >
> > > Thank you once again for your feedback and support. We are glad that our responses have addressed your concerns, and we sincerely appreciate your encouragement. Your comments greatly helped us refine the paper. We will continue to improve the manuscript with care and welcome any further discussions.

---

### Official Review · Reviewer_BSSw · 2025-11-01

**Soundness:** 2
**Presentation:** 3
**Contribution:** 2
**Rating:** 4
**Confidence:** 4

**Summary:**

The GSM8K-V benchmark evaluates vision-language models handling word math problems within visual contexts. It involves converting text-based math problems into cartoon-like images through a specialized pipeline, testing the models on how they reason through the visual data. The study discusses various trials and highlights limitations in current models, underscoring the challenge of accurately combining visual reasoning with textual problem-solving.

**Strengths:**

1. The paper is well-written, with clear figures and detailed explanations that make the work easy to follow.

2. The paper presents a systematic method for converting text-based math problems into visual formats with controlled complexity.

3. The evaluation highlights a significant performance gap between visual and text-based reasoning, providing a target for visual reasoning improvements.

4. The paper identifies key factors like multi-image inputs and caption structures that impact model performance, with the potential to motivate future works.

**Weaknesses:**

1. While the GSM8K-V dataset focuses on educational math problems, these problems may not represent the complexity and variety of real-world mathematical reasoning tasks. The approach of converting problems that do not naturally require visual reasoning into visual formats may not align with how mathematical reasoning occurs in practice.

2. The motivation of the paper remains unclear to me. For the same textual problem, visual problems can be designed with varying levels of difficulty. Introducing irrelevant or distracting information would undoubtedly decrease reasoning accuracy, and I believe this does not provide much valuable insight.

3. The generation of visual scenes relies on predefined templates, which may limit the diversity of scene combinations and fail to reflect the complexity of dynamic and unstructured scenarios.

**Questions:**

1. It would be valuable to categorize errors by their underlying issues, such as spatial reasoning failures, object counting mistakes, or logical inference errors.

2. The authors did not provide explicit guidance on whether the model should analyze the image or solve the problem directly, which could create ambiguity even for humans. I recommend an ablation study examining how explicit instructions about the task focus affect model performance.

---

> ### Author Response · Authors · 2025-11-24
> **Rebuttal [1/2]**
>
> Dear Reviewer BSSw,
> Thank you for your valuable feedback to help us improve our paper. We have revised our paper based on your feedback. We detail our response below and please kindly let us know if our response addresses your concerns.
>
> ---
>
> > **Q1: While the GSM8K-V dataset focuses on educational math problems, these problems may not represent the complexity and variety of real-world mathematical reasoning tasks. The approach of converting problems that do not naturally require visual reasoning into visual formats may not align with how mathematical reasoning occurs in practice.**
>
> A1: Thank you for your question. The existing benchmarks [1-5] tend to focus on geometry or visually structured mathematical content, such as function plots and diagrams. GSM8K-V takes a different perspective. Since GSM8K originally consists of everyday word problems, the textual description can be viewed as an abstraction of real-life scenarios. **GSM8K-V simply restores these descriptions back to realistic visual scenes, which often resemble how mathematical problems naturally appear in daily life.**
>
> From this standpoint, **GSM8K-V examines whether multimodal models can recover quantitative relations when they are not explicitly stated in text, but instead become implicitly embedded in common visual environments.** Rather than emphasizing mathematics itself, the benchmark highlights visual modality, using mathematical questions merely as a vehicle for conveying structured information. This approach provides an anchor for precise evaluation and reflects the way humans frequently encounter problems in real-world situations.
>
> Ref:
>
> [1] lu et al, MathVista: Evaluating mathematical reasoning of foundation models in visual contexts
>
> [2] zhang et al, MathVerse: Does Your Multi-modal LLM Truly See the Diagrams in Visual Math Problems
>
> [3] wang et al, Measuring Multimodal Mathematical Reasoning with MATH-Vision Dataset
>
> [4] wang et al, MathCoder-VL: Bridging Vision and Code for Enhanced Multimodal Mathematical Reasoning
>
> [5] rasheed et al, VideoMathQA: Benchmarking Mathematical Reasoning via Multimodal Understanding in Videos
>
> > **Q2: The motivation of the paper remains unclear to me. For the same textual problem, visual problems can be designed with varying levels of difficulty. Introducing irrelevant or distracting information would undoubtedly decrease reasoning accuracy, and I believe this does not provide much valuable insight.**
>
> A2: Thank you for your question. **GSM8K-V reflects how mathematical problems naturally appear in daily life by placing them in realistic visual scenes, rather than keeping them in purely textual form. A key motivation is to examine whether multimodal models can recover quantitative information when content that is explicit in text becomes embedded in realistic visual scenes.** This change reflects a critical difference between textual and visual modalities. In text, semantic information is directly expressed through symbols and is immediately accessible. In images, the same content is conveyed through perceptual cues. From this perspective, GSM8K-V exposes a clear gap between understanding textual information and reasoning over visually grounded content.
>
> Regarding interference, irrelevant details indeed cause some performance degradation, but they are not the main reason for the substantial drop in accuracy. A detailed error analysis shows that the dominant failures arise from the model’s difficulty in recovering implicitly encoded visual information and various perceptual mistakes. **GSM8K-V introduces interference to make the original text-based math problems better reflect realistic visual scenarios.** We have revised the paper and put these details in Section 5.5 and Appendix E.
>
> > **Q3: The generation of visual scenes relies on predefined templates, which may limit the diversity of scene combinations and fail to reflect the complexity of dynamic and unstructured scenarios.**
>
> A3: Thank you for your question. The predefined visual templates operate only at the level of individual math information (e.g., Beck has 5 apples, Beck eats 5 apples). **Scene combinations are not template-assembled but are constructed by GPT-4.1 following the guiding principles (contextual grouping, final isolation, atomic fidelity, details in Sec 3.1) and are further checked by human annotators.** Thus the final multi-scene sequences result from free-form generative composition rather than rigid template binding. Structural organization is imposed solely to ensure successful image rendering and stable cross-scene consistency.

---

> ### Author Response · Authors · 2025-11-24
> **Rebuttal [2/2]**
>
> > **Q4: It would be valuable to categorize errors by their underlying issues, such as spatial reasoning failures, object counting mistakes, or logical inference errors.**
>
> A4: Thank you for your question. To address the question, we reanalyse four major error types in GSM8K-V, based on a detailed examination of the questions and the corresponding model errors.
> - **Perception Calculation Error:** Models often misperceive counts or confuse object categories, leading to cascading calculation mistakes. This error frequently flips the outcome of arithmetic reasoning and underscore a key weakness in visual perception fused with quantitative binding.
> - **Implicit Visual Inference Error:** This error occurs when key relations are conveyed through implicit visual cues instead of explicit depiction. Models often overlook such dependencies and treat them as isolated visual details rather than components of a coherent relational structure. These failures show a weakness in inferring relations that are visually suggested but not directly stated.
> - **Instrument-Reading Error:** This error arises when models fail to correctly interpret everyday measurement instruments that encode numerical values.
> - **Other Error:** This category includes all cases outside the three types above. It mainly consists of visual confusion and OCR-related character recognition mistakes.
>
> Based on these four types, we analyze the error distribution of Qwen2.5-VL-72B and GPT-4o. **Perception Calculation Error and Implicit Visual Inference Error are the dominant sources of failure.** They account for 34% and 28% of errors in Qwen2.5-VL-72B, and 30% and 26% in GPT-4o. This indicates that current visual language models still suffer from deficits in perception, which aligns with widely discussed limitations, and also in implicit visual reasoning. We have revised the paper and put these details in Section 5.5 and Appendix E. Thank you again for the helpful suggestion.
>
> > **Q5: The authors did not provide explicit guidance on whether the model should analyze the image or solve the problem directly, which could create ambiguity even for humans. I recommend an ablation study examining how explicit instructions about the task focus affect model performance.**
>
> A5: We thank the reviewer for suggesting an ablation on prompt design. We had considered this point earlier and conducted a corresponding ablation experiment, reported in Section 5.1 Ablation on the format of input query. Due to space limitations in the main text, the detailed results were placed in Appendix B.2.2 Implicit and explicit. In this ablation, we compare two input modes: an implicit mode, where the model is given only the scene images, and an explicit mode, where the scene images and a final masked question without revealing the mathematical content are provided. Across 19 evaluated models, we observe no substantial performance difference between the two settings, with variations within approximately one point. **This indicates that the performance degradation does not stem from difficulty in identifying what the problem is, but rather from the inherent challenge of the visual reasoning task itself.**
>
> ---
>
> Thank you again for the thorough review. We think that our rebuttal can address your main concern, if you have any further questions, we would be happy to answer them.
>
> Sincerely,
> GSM8K-V Authors

---

### Author Response · Authors · 2025-12-02
**General Response**

Dear Reviewers, AC, SAC, and PC,

We sincerely appreciate the time and dedication you invested in reviewing our work. Your thoughtful comments and constructive suggestions greatly helped us refine the clarity, motivation, and contributions of the GSM8K-V benchmark. In the following response, we summarize the key elements of our design, motivation, comparisons, and update details in the revised version of the paper.

> **Brief introduction for GSM8K-V**

GSM8K-V builds on the nature of GSM8K: the original problems are everyday math scenarios, and the text is only an abstraction of real situations. GSM8K-V restores these problems to visual scenes, reflecting how such reasoning typically appears in daily life. We evaluate a wide range of open-source and close-source models on the proposed GSM8K-V benchmark. Our results reveal that, although existing VLMs have achieved nearly saturated performance on the text-based GSM8K, there remains substantial room for improvement on the purely visual GSM8K-V.



> **Motivation of GSM8K-V**

The motivation of GSM8K-V is to **construct a benchmark that evaluates mathematical reasoning as it naturally occurs in everyday visual contexts**. The conversion to visual format enables controlled comparison between text-based reasoning and visual-based reasoning under the same semantic content, allowing us to isolate whether failures come from inability in visual perception, visual implicit reasoning or mathematical reasoning ability. To summarize, the goal is not to make the task harder by removing text, but to see whether models can solve the same problems when they arise visually in daily life, as humans often encounter them.



> **Comparison between GSM8K-V and current visual mathematical benchmark**

Existing benchmarks [1-5] focus on structured diagrams or geometry-based layouts. Their reasoning process relies on recognizing visual content and then converting it into symbolic text for calculation.  And current benchmarks (e.g., MathVista [1], VideoMathQA [5]) **rarely evaluate whether models can extract mathematically relevant information directly from everyday visual scenes**, where key cues are implicitly embedded in natural contexts(i.e., everyday visual reasoning). They primarily focus on domain-specific skills and structured knowledge, rather than everyday reasoning found in real-life situations. Table R1 summarizes the difference between GSM8K-V and other benchmark.

**Table R1: Comparison of visual mathematicl benchmarks**
| Benchmark          | Input           | Evaluated Abilities                                                                                                            | Target Scenario                                 |
| ------------------ | --------------- | ------------------------------------------------------------------------------------------------------------------------------ | ----------------------------------------------- |
| MathVista [1]      | Single-image    | Geometry & Algebra, Perception-based reasoning                                                                                 | Classroom-style visual math problems            |
| MathVision [3-4]   | Single-image   | Geometry, Diagram understanding, Logical reasoning                                                                             | Classroom-style structured math tasks           |
| MathVerse [2]      | Single-image    | Symbolic reasoning after visual perception, Diagram interpretation                                                             | Classroom-style math-worksheet problems         |
| VideoMathQA [5]    | Video           | Temporal reasoning, Cross-frame perception, Elementary logical reasoning                                                       | Instructional video-based math tasks            |
| **GSM8K-V (Ours)** | **Multi-image** | **Perception & Counting, Visual implicit reasoning(Cross-scene integration, Partial observability, Contextual reasoning), Non-domain-specific logic reasoning** | **Real-world math word problems in daily life** |

---

---

> ### Author Response · Authors · 2025-12-02
> **General Response**
>
> To verify that GSM8K-V targets at different evaluation ability, we conduct performance comparion experiment based on VL-Rethinker-7B [7] in Table R2.
>
>
> **Table R2: Performance comparison between Qwen2.5-VL-7B-Instruct and VL-Rethinker-7B across multiple benchmarks.**
>
> | Model                      | MathVista (testmini) | MathVerse (testmini) | MathVision (test) | GSM8K-V  |
> | --- | :---: | :---: | :---: | :---: |
> | Qwen2.5-VL-7B-Instruct [6] | 68.2                 | 46.3                 | 25.1              | 15.16    |
> | VL-Rethinker-7B [7]        | 74.9                 | 54.2                 | 32.3              | 14.56    |
> | **Δ**                      | **+6.7**             | **+7.9**             | **+7.2**          | **−0.6** |
>
> As shown in Table R2, VL-Rethinker-7B is derived from Qwen2.5-VL-7B-Instruct and further trained on ViRL39K [7] (38,870 verifiable QAs for vlm RL training, ~ 80% math type).  From Table R3, RL training on ViRL39K significantly improves performance on MathVista, MathVerse, MathVision(+6.7, +7.9, +7.2). However, performance drops on GSM8K-V (−0.6). **This result indicates that GSM8K-V represents a distinct task type rather than an extension of existing benchmarks.**
>
> Reference:
> [1] Lu et al, MathVista: Evaluating mathematical reasoning of foundation models in visual contexts
> [2] Zhang et al, MathVerse: Does Your Multi-modal LLM Truly See the Diagrams in Visual Math Problems
> [3] Wang et al, Measuring Multimodal Mathematical Reasoning with MATH-Vision Dataset
> [4] Wang et al, MathCoder-VL: Bridging Vision and Code for Enhanced Multimodal Mathematical Reasoning
> [5] Rasheed et al, VideoMathQA: Benchmarking Mathematical Reasoning via Multimodal Understanding in Videos
> [6] Qwen Team, Qwen2.5-VL Technical Report
> [7] wang et al, VL-Rethinker: Incentivizing Self-Reflection of Vision-Language Models with Reinforcement Learning
>
>
> > **Revision update details**
>
> - Section 5.2 ABLATION ON MULTI-IMAGE INPUTS
> To assess how different organizational formats influence model performance, we add additional experiment on Single-Image,Single-Image(r*), Multi-Image, and Multi-Image(r*), where r* denotes a random permutation of scene order applied consistently across both single-image and multi-image conditions. The results (Table 3) show that models with weaker cross-scene integration capabilities are particularly sensitive to perturbations in scene organization. Appendix B.4 further update illustrates details.
>
> - Section 5.3 SENSITIVITY ON IMAGE STYLE
> To verify the generalizability of cartoon-style illustrations, we extend our ablation study to include *Giphli-style* with softer tones and painterly textures and *Real-world* style with photorealistic.The result (Table4) confirms that stylistic variation minimally impacts reasoning and demonstrates robustness to photorealistic rendering, validating cross-style generalization. Appendix B.5 provides updated side-by-side illustrations.
>
> - Section 5.4 MODALITY VALID
> To evaluates whether GSM8K-V genuinely requires visual reasoning (i.e., can not be solved by ocr and textual reasoning) by comparing *text-only*, *image-only*, *OCR*, *OCR plus Image* and *caption* inputs. The updated result shown in Table 5 verify that key semantic in GSM8K-V is often embedded implicitly across the multi visual scenes rather than directly text expressed. Appendix B.3 provides updated illustrations.
>
> - Section 5.5 ERROR ANALYSIS
> Based on manual examination of model outputs, we conducted a detailed error analysis to clearly separate different failure sources. We classify errors into four diagnostic categories:
>     - Perception Calculation Error: Miscounting objects or confusing categories leads to incorrect numerical grounding and subsequent calculation mistakes.
>     - Implicit Visual Inference Error: Key relations appear implicitly in visual context, but models often treat them as irrelevant details, failing to form coherent relational reasoning.
>     - Instrument-Reading Error: Numerical values are encoded in everyday measurement instruments, and models sometimes misinterpret them.
>     - Other Error: Includes cases mainly related to OCR-type character mistakes and general visual confusion.
>
>     We update the detailed examples and error distribution in Appendix E.
>
> - Appendix G Comparison between GSM8K-V and other benchmark
> We list difference between GSM8K and existing visual mathematical benchmark (e.g., MathVista, MathVision, VideoMathQA). We conducted a comparison experiment using VL-Rethinker-7B across these benchmarks.
>
> Once again, we sincerely thank all reviewers for their insightful feedback and invested time!
>
> Sincerely,
> GSM8K-V Authors

---

### Author Response · Authors · 2025-12-02
**Summary of Rebuttal**

Dear Reviewers, AC, SAC, and PC,

We sincerely appreciate the time and effort you devoted to reviewing our work, as well as the insightful feedback provided. The constructive comments greatly helped us strengthen the paper. Below, we summarize the key points of progress.

---
We are encouraged to see that several reviewers recognized the essential contributions of GSM8K-V, including:

- **Novel and well-motivated benchmark:** GSM8K-V is the first purely visual and multi-image counterpart of GSM8K, enabling one-to-one comparison between text-based and visual reasoning under identical semantics. And the benchmark exposes the modality gap between textual reasoning and everyday visual reasoning, and enables isolating failures from visual perception, visual implicit reasoning, and mathematical reasoning. (Reviewer `bjKz`, `F6kb`)
- **Comprehensive evaluation and ablation:** Ablations, error categorization, and style sensitivity analysis provide reproducible and actionable insights for diagnosing VLM limitations. (Reviewer `bjKz`, `F6kb`, `GZam`)
- **Well-designed construction pipeline:** The three-stage framework (text decomposition → scene generation → human verification) ensures semantic fidelity and consistency. (Reviewer `bjKz`, `F6kb`)
- **Writing and presentation are clear, with illustrative examples aiding understanding.** (Reviewer `BSSw`, `bjKz`)

---


During the rebuttal phase, we conducted additional experiments supplemented relevant analyses in the revision to address the reviewers' concerns:

- We list difference between GSM8K and existing visual mathematical benchmark (e.g., MathVista, MathVision, VideoMathQA). We conducted a comparison experiment using VL-Rethinker-7B, which improves significantly on MathVista, MathVerse, and MathVision, but achieves no improvement on GSM8K-V (−0.6). This confirms that **GSM8K-V represents a distinct evaluation dimension rather than an extension of existing benchmarks.** These results are in Appendix G in our revised paper. (Reviewer `GZam`)
- We added experiments using OCR-only, OCR+Image, and MinerU2.5 / PaddleOCR-VL models for modality valid. The results show that **GSM8K-V cannot be solved by text extraction alone, confirming that genuine visual reasoning is required.** Details are provided in Appendix B.3. (Reviewer `GZam`, `F6kb`)
- We extended the **style sensitivity analysis** to include Giphli-style and real-world photorealistic rendering. Performance differed by less than three points across all styles, demonstrating robustness and cross-style generalization. (Reviewer `bjKz`, `F6kb`)
- To investigate whether performance drop is caused by multi-image structure, we added Single-Image, Single-Image (random order), Multi-Image, and Multi-Image (random order) settings. Results indicate that weaker models are sensitive to scene order, while stronger models are primarily challenged by implicit visual semantics, rather than organizational format alone. (Reviewer `F6kb`, `BSSw`)

---

We also clarified several key points to address the reviewers' concerns:


- We explained that the motivation of GSM8K-V is to **construct a benchmark that evaluates mathematical reasoning as it naturally occurs in everyday visual contexts**. The conversion to visual format enables controlled comparison between text-based reasoning and visual-based reasoning under the same semantic content, allowing us to isolate whether failures come from inability in visual perception, visual implicit reasoning or mathematical reasoning ability. (Reviewer `BSSw`, `GZam`)
- We introduced four diagnostic error categories: Perception Calculation Error, Implicit Visual Inference Error, Instrument-Reading Error, and Other—and analyzed their distributions. The dominant sources of failure are perception and implicit visual reasoning, not OCR-related mistakes. Details were updated in Section 5.5 and Appendix E. (Reviewer `F6kb`, `BSSw`)

---
Regarding reviewer feedback during the discussion phase:

- Reviewer `bjKz` stated that all concerns addressed, this is a fantastic research and deserves to be published, and explicitly increased their score in favor of acceptance.
- Due to current ICLR policy, we were unable to engage in discussion with Reviewer `GZam`, Reviewer `F6kb` and Reviewer `BSSw`.

---
Once again, we sincerely thank all reviewers for their recognition and constructive suggestions. We believe that our responses and revisions have effectively addressed the raised concerns, and we hope the committee (reviewers, ACs, SACs, PCs) will take them into account in the next stages.

Sincerely,
GSM8K-V Authors

---

### Note · Program_Chairs · 2026-01-17
**Submission Desk Rejected by Program Chairs**

The following references in this submission do not refer to real documents and/or have major errors in bibliographic information:

 Yuxuan Wang et al. MINT-CoT: Enabling interleaved visual tokens in multimodal chain-of-thought. arXiv preprint arXiv:2506.05331, 2025e. URL https://arxiv.org/abs/2506.05331.
Yuwei Wang et al. Reinforcement Learning with Verifiable Rewards (RLVR) improves mathematical reasoning. arXiv preprint arXiv:2504.20571, 2025d. URL https://arxiv.org/abs/ 2504.2057